# THE POWER OF THE SENSES: GENERALIZABLE MANIPULATION FROM VISION AND TOUCH THROUGH MASKED MULTIMODAL LEARNING

## ABSTRACT

Humans rely on the synergy of their senses for most essential tasks. For tasks requiring object manipulation, we seamlessly and effectively exploit the complementarity of our senses of vision and touch. This paper draws inspiration from such capabilities and aims to find a systematic approach to fuse visual and tactile information in a reinforcement learning setting. We propose **M**asked **M**ulti**m**odal **L**earning (**M3L**), which jointly learns a policy and visual-tactile representations based on masked autoencoding. The representations jointly learned from vision and touch improve sample efficiency, and unlock generalization capabilities beyond those achievable through each of the senses separately. Remarkably, representations learned in a multimodal setting also benefit vision-only policies at test time. We evaluate M3L on three simulated environments with both visual and tactile observations: robotic insertion, door opening, and dexterous in-hand manipulation, demonstrating the benefits of learning a multimodal policy. Videos of the experiments are available at `https://m3l.site`. Code will be released upon acceptance.

## 1 INTRODUCTION

Humans are capable of exploiting the synergies and complementarities of their senses (Blake et al., 2004; Zhou et al., 2010; Macaluso & Driver, 2001). For example, when grasping an object, we fully rely on our sense of vision at first, since no physical feedback is available until contact is made. Once the object has been reached, visual feedback becomes partly or fully occluded by the human hand. Thus, vision-based reasoning is naturally replaced by rich touch feedback. Human reasoning and decision-making present uncountable similar examples, where different sensory modalities seamlessly cooperate with each other.

However, in robotic manipulation, vision and touch have mostly been studied independently, mainly due to the delayed development of tactile sensors compared to the widespread availability of high-performance visual sensing. While vision-based manipulation research has shown impressive achievements through modern machine learning approaches (Zhang et al., 2015; Kalashnikov et al., 2018), incorporating contact feedback with vision is crucial to broaden the capabilities of robotic manipulation, e.g., dealing with visual occlusion, manipulating fragile objects, and improving accuracy. Yet, a large part of touch-based manipulation research has so far focused on showcasing the potential of new high-resolution tactile sensors (Hogan et al., 2020; She et al., 2021), often limited to proof-of-concepts based on the assumption that visual sensing is unavailable.

In this paper, we propose Masked Multimodal Learning (M3L), which leverages both visual and tactile sensing modalities by systematically fusing them for manipulation tasks. Specifically, we focus on sample efficiency and generalization of reinforcement learning (RL) via multimodal representations extracted across vision and touch. To acquire such generalizable multimodal representations, we use a multimodal masked autoencoder (MAE) (He et al., 2022) that learns to extract condensed representations by optimizing a reconstruction loss based on raw visual and tactile observations, while simultaneously optimizing a policy that is conditioned on such representations.

We show that the multimodal representations learned through M3L result in better sample efficiency and stronger generalization capabilities compared to settings that treat each modality separately. In

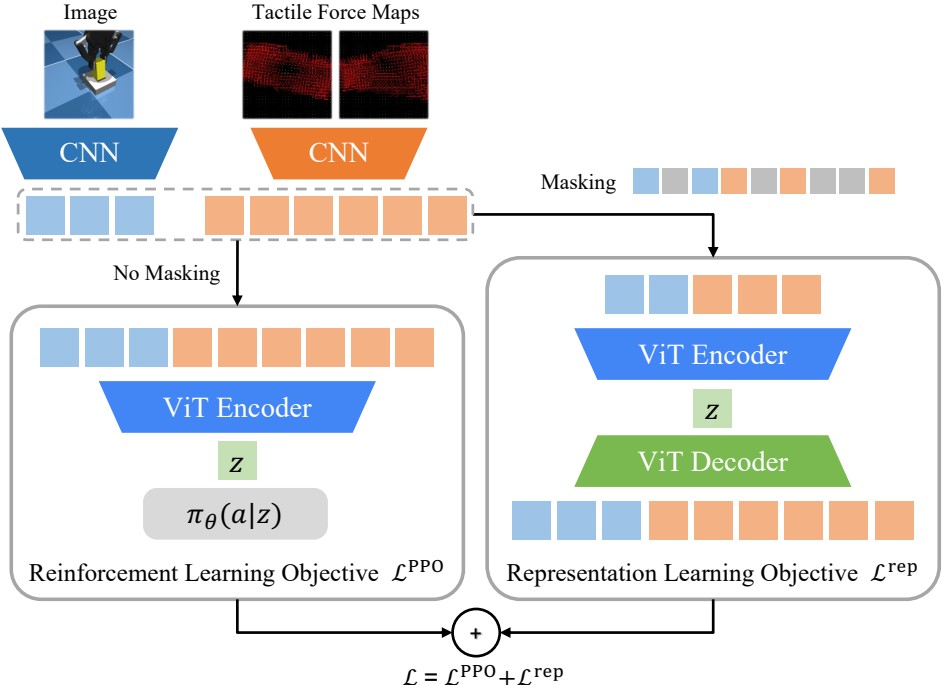

Figure 1: Masked Multimodal Learning (M3L) framework. M3L simultaneously optimizes a representation learning loss and a reinforcement learning objective. A policy is trained using Proximal Policy Optimization (PPO) (Schulman et al., 2017), conditioned on multimodal representations learned through a masked autoencoder (MAE) (He et al., 2022). By attending to each other within a unified vision transformer (ViT) encoder, visual and tactile data provide representations that lead to more generalizable policy learning. Note that the ViT encoders used for representation and policy learning share weights with each other.

particular, M3L demonstrates better zero-shot generalization to unseen objects and variations of the task scene, exploiting the representation power provided by multimodal reasoning. Moreover, we observe that the aforementioned generalization capabilities are substantially retained even if the representation encoder, trained with multimodal data, is deployed to a vision-only policy. This suggests that the generalization benefits of touch are strongly intertwined with how the policy learns its representations, offering the possibility to trade off a limited loss of performance with the additional complications of using touch sensors for robot deployment in the real world.

## 2 RELATED WORK

**Reinforcement Learning for Manipulation.** The growing application of computer vision in robotics has enabled robotic manipulation policies trained from raw pixel observations through reinforcement learning (RL) (Zhang et al., 2015; Kalashnikov et al., 2018). However, a vision-based policy struggles with occlusions and only enables a delayed response for contact-rich tasks.

Thanks to the recent advances in the development of high-resolution tactile sensors (Yuan et al., 2017; Ward-Cherrier et al., 2018; Sferrazza & D'Andrea, 2019; Park et al., 2022; Lambeta et al., 2020; Bhirangi et al., 2021), touch-based manipulation has tried to address the visual occlusion problem and enable reactive contact-rich manipulation with local information from tactile sensors. Common examples are in-hand manipulation (Melnik et al., 2021; Yin et al., 2023), Braille alphabet reading (Church et al., 2020), pendulum swingups using learned feedforward (Wang et al., 2020) or feedback policies (Bi et al., 2021), or peg-hole insertion using primitive trajectories (Dong & Rodriguez, 2019; Dong et al., 2021) and task-specific controllers (Kim & Rodriguez, 2022). However, these approaches lack the use of visual information (Lin et al., 2022; Huang et al., 2019), which is often required for global reasoning about the task.

The combination of vision and contact feedback has been investigated in various settings, such as model predictive control (Fazeli et al., 2019) and behavioral cloning (Li et al., 2022). More recently, various efforts have also been made in the context of model-free RL (Pecyna et al., 2022; Wu et al., 2019; Qi et al., 2023), which is the focus of our work and promises to learn control policies without the need for models of the system at hand or expert demonstrations. An end-to-end RL strategy from visual and tactile data, pre-processed through two separate neural networks, was shown in Hansen et al. (2022) on the Robosuite benchmark, where tactile signals were obtained through an approximation of the object depth map. In our work, rather than learning end-to-end, we focus on sample efficiency and generalization through a self-supervised representation learning objective.

**Representation Learning for Manipulation.** Representation learning has played a key role in reducing sample complexity when applying RL to high-dimensional observation spaces (Zhan et al., 2022; Seo et al., 2023a; Xiao et al., 2022; Radosavovic et al., 2023). In this context, several studies have focused on extracting condensed representations from tactile inputs (Chebotar et al., 2014; Van Hoof et al., 2016; Sutanto et al., 2019). A notable exception is Lee et al. (2019), where a force-torque sensor was used in combination with vision, and a self-supervised learning architecture was found to improve sample efficiency compared to learning from raw data.

More recently, representation learning has been applied from visual and tactile data in contexts different from RL. Specifically, Guzey et al. (2023b) trained tactile and visual encoders in a self-supervised manner, and exploited the extracted representations via imitation and residual learning (Guzey et al., 2023a) for manipulation tasks. In Kerr et al. (2022), a general perception module was proposed by training two separate (vision and touch) encoders using a contrastive approach. Our work differs in that we focus on fusing visual and high-resolution tactile inputs through a joint encoder that learns interrelations between the two modalities, particularly enhancing generalization capabilities. We focus on a specific class of representation learning algorithms, based on masked autoencoding (He et al., 2022).

**Masked Autoencoders for Manipulation.** The idea of learning representations by reconstructing the masked parts of images (Vincent et al., 2010; Pathak et al., 2016) has recently been scaled up inspired by the idea of masked language modeling in the language domain (Devlin et al., 2019) and the introduction of the Transformer architecture (Vaswani et al., 2017). Notably, He et al. (2022) introduced Masked Autoencoders (MAE) that randomly mask patches of images and reconstruct the masked parts based on the vision transformer (ViT) architecture (Dosovitskiy et al., 2021). Recent works have demonstrated that MAE representations can be useful for learning manipulation policies from pixel observations (Xiao et al., 2022; Radosavovic et al., 2023; Seo et al., 2023b;a; Liu et al., 2022). In particular, the works closely related to ours have proposed to learn joint representations with MAEs and utilize it for robotic manipulation. For instance, Liu et al. (2022) utilized frozen representations from a pre-trained vision-language multimodal MAE (Geng et al., 2022) for learning instruction-following manipulation policies. Seo et al. (2023b) trained an MAE with visual observations from multiple cameras and utilized it for RL. In this context, our work further demonstrates that learning joint vision-touch representations by training a multimodal MAE improves the sample efficiency and generalization of robotic manipulation policies.

## 3 BACKGROUND

**Reinforcement Learning (RL).** We formulate the problem as a Markov decision process (MDP) (Sutton & Barto, 2018), which is defined as a tuple $(\mathcal{S}, \mathcal{A}, p, r, \gamma)$. Here, $\mathcal{S}$ denotes the state space, $\mathcal{A}$ denotes the action space, $p(s_{t+1}|s_t, a_t)$ is the transition dynamics, $r$ is the extrinsic reward function $r_t = r(s_t, a_t)$, and $\gamma \in [0, 1]$ is the discount factor. The goal of RL is to train a policy $\pi$ to maximize the expected return. Our approach is compatible with any RL algorithm, but here we use Proximal Policy Optimization (PPO) (Schulman et al., 2017) as our underlying RL algorithm due to its simplicity and scalability with parallel environments (Makoviychuk et al., 2021). We refer to Appendix B for more details about PPO.

**Masked Autoencoding for Representation Learning.** Masked autoencoding (He et al., 2022) is a self-supervised learning method that learns image representations by reconstructing the masked parts of images given the unmasked parts. Specifically, a masked autoencoder (MAE) first divides the

images into non-overlapping square patches and adds positional embeddings (Vaswani et al., 2017) to the patches. Then, it randomly masks the patches, and a vision transformer (ViT) (Dosovitskiy et al., 2021) encoder computes the visual embeddings of the remaining (unmasked) patches through a series of transformer layers (Vaswani et al., 2017). Because the ViT encoder only processes a small subset of full patches (e.g., typically 25%), training becomes more compute-efficient and scalable. For decoding, learnable mask tokens (Devlin et al., 2019) are concatenated with the unmasked patch representations and the positional embeddings are added in order to represent the position of masked patches to be reconstructed. Finally, a ViT decoder takes the concatenated inputs and outputs predicted pixel patches. All model parameters are updated to minimize the mean squared error (MSE) between the predicted pixel patches and the original patches.

**High-resolution Tactile Sensing Measurements.**
Modern high-resolution tactile sensors (Yamaguchi & Atkeson, 2019; Abad & Ranasinghe, 2020; Shimonomura, 2019) may provide touch feedback in the form of spatially distributed quantities, such as deformation fields, strain fields, and force maps. In particular, force maps have been shown to be measurable in the real world through a variety of tactile sensors (Sferrazza & D'Andrea, 2022; Zhang et al., 2022; Ma et al., 2019), are readily available through physics simulators (e.g., MuJoCo touch grid or the approach presented in Xu et al. (2023)), and have been demonstrated as a valid abstraction to achieve successful sim-to-real transfer in highly dynamic manipulation tasks (Bi et al., 2021). The elements comprising a force map are generally denoted as "taxels", i.e., the

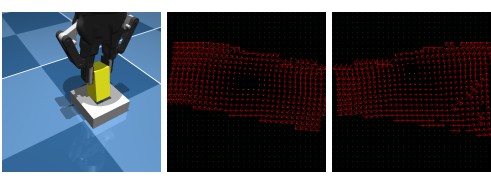

(a) Image   (b) Tactile left   (c) Tactile right

Figure 2: Visualization of observations from the tactile insertion environment: (a) $64 \times 64$ visual input and (b, c) two $32 \times 32$ tactile inputs (taxels), where the color of the arrows indicates pressure (red means high pressure) and the direction indicates shear, following the convention in Xu et al. (2023).

tactile dual of pixels. Such maps are often represented in a similar way as images, that is, in a `channels × height × width` form, where the channels are usually the three components: two for shear and one for pressure of the contact force, as shown in Figure 2.

## 4 METHOD

In this section, we present **M**asked **M**ulti**m**odal **L**earning (**M3L**), a representation learning technique for reinforcement learning that targets robotic manipulation systems provided with vision and high-resolution touch. Specifically, M3L learns a policy conditioned on multimodal representations, which are extracted from visual and tactile data through a shared representation encoder. As illustrated in Figure 1, the M3L representations are trained by optimizing at the same time representation learning and reinforcement learning objectives:

$$\mathcal{L} = \mathcal{L}^{\texttt{rep}} + \mathcal{L}^{\texttt{PPO}}, \tag{1}$$

where $\mathcal{L}^{\texttt{rep}}$ is the multimodal representation learning objective (Section 4.1) and $\mathcal{L}^{\texttt{PPO}}$ is PPO's reinforcement learning objective (Section 4.2).

### 4.1 REPRESENTATION LEARNING

M3L achieves multimodal representation learning by using both image and tactile data to update an MAE that learns to reconstruct both pixels and taxels at the same time. This can be written as following:

$$\mathcal{L}^{\texttt{rep}} = \text{MSE}^{\texttt{pixels}} + \beta_T \cdot \text{MSE}^{\texttt{taxels}}, \tag{2}$$

where $\beta_T$ is a hyperparameter that balances the two MSE losses for vision and touch.

Note that as opposed to other representation learning approaches, such as contrastive learning, MAEs do not need discovering new data augmentations and invariances to design positives and negatives. On the other hand, patching and reconstructing in MAEs seamlessly apply to tactile data. In addition, the transformer architecture and the masking scheme support input sequences of variable length and facilitate design strategies particularly suited for multimodal data, e.g., vision and touch.

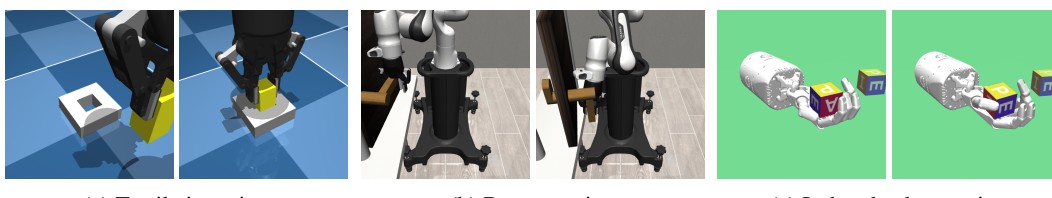

(a) Tactile insertion     (b) Door opening     (c) In-hand cube rotation

Figure 3: We evaluate M3L on three simulated environments: (a) Tactile insertion, (b) Door opening, and (c) In-hand cube rotation. For each task, the left images represent initial configurations and the right images show task completion.

We list below the most relevant implementation details of our representation learning framework.

**Early Convolutions.** The MAE encoder has two preprocessing convolutional neural networks (CNNs) that compute convolutional features from pixels and taxels, respectively. Such convolutional features are then masked in place of the raw input patches. These early convolution layers help capturing small details in reconstruction (Seo et al., 2023a).

**Positional and Modality Embeddings.** As standard for transformers, we add 2D sin-cos positional embeddings (Chen et al., 2021) to both the encoder and decoder features. In addition, we also add learnable 1D modality embeddings representing either visual or tactile streams, following the implementation of Geng et al. (2022) for vision and language.

**Reconstruction Pipeline.** Convolutional features are computed as described above for $k$ frames concatenated over time. In particular, we concatenate the frames in the channel dimension (e.g., concatenation of RGB images results in a $3k$-channel tensor). Frame stacking turned out to be crucial for success on the environments considered in our experiments (see Section 6). We then uniformly mask across visual and tactile features. Finally, we feed the unmasked convolutional features from both vision and touch into the MAE for reconstruction, so that the ViT encoder can attend to both modalities.

## 4.2 POLICY LEARNING WITH M3L

The policy learning closely follows PPO with the exception of how the observations are extracted from the raw input data. At each time step, the image and tactile data are fed into the preprocessing CNNs. The CNN features are then added to the positional and modality embeddings and processed through the MAE encoder, without applying any masking. The extracted multimodal embeddings are then provided to the actor and critic networks. Each of these consist of a transformer layer that processes the embeddings and aggregates them through a global average pooling layer, and a multilayer perceptron (MLP) that outputs either the value (for the critic) or the mean of the action distribution (for the actor). Note that the gradients computed through the PPO loss are also propagated up to the MAE encoder and the CNNs. As a result, the CNNs and MAE encoder are updated to simultaneously optimize both representation and task learning.

The overview of M3L is illustrated in Figure 1 and detailed in Appendix A. More details about PPO can be found in Appendix B.

## 5 SIMULATION ENVIRONMENTS

We perform our experiments in three simulated environments using MuJoCo (Todorov et al., 2012)'s touch-grid sensor plugin, which aggregates contact forces into taxels. To the best of our knowledge, the following are the first examples where high-resolution force fields have been included in a MuJoCo robotics environment, which can also seamlessly render visual information. We will make our environments public for reproducibility and further research in visual-tactile manipulation.

## 5.1 TACTILE INSERTION

The tactile insertion environment consists of a peg object and a target frame where the peg can be tightly inserted, and the Menagerie's (MuJoCo Menagerie Contributors, 2022) Robotiq 2F-85 parallel-jaw gripper model, as shown in Figure 3a. Each finger has the silicone pad modeled as a rectangular prism (`box` geometry in MuJoCo). In MuJoCo, contact sensing with a box primitive is computed only at the four vertices. Therefore, we split the collision mesh of the box into a grid of smaller boxes, to increase the number of candidate contact points, and consequently the effective resolution of the force map. The resulting tactile observation is in the form of two 32×32 taxel maps (one per finger). Each taxel corresponds to a 3D force vector, which represents both shear and pressure forces, as shown in Figure 2. In addition, the observation also includes a $64 \times 64$ image.

Each episode starts with the peg held between the gripper fingers with a randomized initial position of the gripper in the 3D space. We also randomize the shape of the peg (see all 18 peg shapes in Appendix D), the shape of the target frame (square or circle), and the target hole location. The control inputs to the system are the 3D coordinates of the floating gripper, while we fix both the gripping force and gripper rotation. The task comprises 300 steps and is considered to be solved once the object position is within a small threshold from the target position. We use a dense reward, which is the negative distance between the peg and the target position, as well as a sparse task completion reward of 1000.

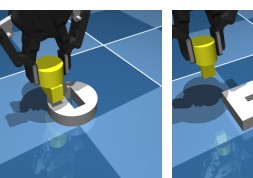 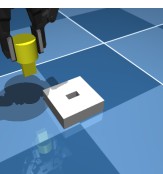 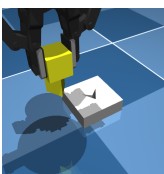

(a) Training peg   (b) Rectangle peg   (c) V-shape peg

Figure 4: We use 18 different training peg shapes, and 2 novel pegs (rectangle and V-shape) to test generalization on the tactile insertion task.

Note that while Dong et al. (2021); Xu et al. (2023) also address the insertion task with tactile information, they heavily rely on prior knowledge (e.g. initial estimate of the insertion region and open-loop insertion trajectory) and only learn to correct errors online using tactile information. On the other hand, our goal is to benchmark a general RL approach that can utilize vision and touch together to generate raw control actions, without requiring any prior information.

## 5.2 DOOR OPENING

The door opening task from Robosuite (Zhu et al., 2020) requires to open a locked door by turning the door handle and then pulling the door with a Franka robot arm and a Robotiq 2F-85 gripper, as shown in Figure 3b. We extend this environment by adding tactile sensors to gripper fingers as in the tactile insertion environment. The observation space comprises a $64 \times 64$ camera image and two $32 \times 32$ tactile maps. The action space consists of 3D delta end-effector position and rotation. Note that the gripper is always closed, holding the door handle.

To make the exploration problem easier and focus on generalizable skill learning, we provide additional dense rewards for opening the door and a sparse success reward of 300 when the door is opened. We initialize the robot to hold the door handle and the position of the door is fixed at $(0.07, 0.00)$. Each episode lasts for 300 steps but terminates when the door is opened or the gripper detaches from the door handle. To test generalization capability, we randomly initialize the door position, $x \sim [0.06, 0.10], y \sim [-0.01, 0.01]$ and use $10\times$ higher friction and damping coefficients for hinges of both the door and door handle during testing.

## 5.3 IN-HAND ROTATION

The in-hand cube rotation task is based on the in-hand block reorientation environment (Plappert et al., 2018) provided through Gymnasium-Robotics. The environment relies on a Shadow Robot Dexterous Hand with 20D actions. We augment the visual observation with high-resolution force maps. Specifically, we add $3 \times 3$ force maps to each of the finger phalanges and to the palm of the hand. Through the use of zero-padding, we rearrange such force readings into a $32 \times 32$ map, as illustrated in the Appendix, Figure 9.

The task consists in reorienting a colored cube to a predefined configuration, overlaid next to the actual hand-cube system (see Figure 3c). We use a reward of 100 when the cube is within a threshold

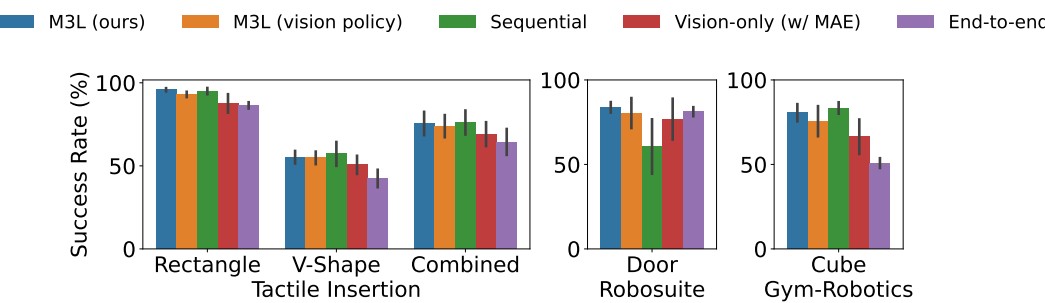

Figure 5: Zero-shot generalization experiments on the three tasks. The bar plots show mean and standard error on 5 seeds, with 25 episodes run after training for the 4 last checkpoints on each seed.

from the target, in addition to the dense reward implemented in the original environment. To test generalization, we double the mass of the cube and slightly perturb the camera pose, and attempt the same reorientation task.

We found that this task requires a higher level of accuracy in the representations (e.g., to properly detect the different faces of the cube) compared to the previous two. For this reason, rather than directly optimizing the sum of two objectives in Equation (1), we perform a reinforcement learning gradient descent step every $n$ representation learning gradient steps. In particular, given an RL batch size $B$, we split this batch in $n$ chunks for the representation learning phase and then use the full batch for the reinforcement learning phase.

We present additional in-hand rotation tasks with different objects, e.g., an egg and a pen, in the appendix.

## 6  EXPERIMENTS

In this section, we study the advantages of M3L in visual-tactile manipulation compared to baselines, and explain our design choices. In particular, we aim to answer the following questions:

- Does our multimodal approach improve generalization when manipulating unseen objects or dealing with scene variations?
- Is the representation learning loss beneficial compared to training the same architecture end-to-end via PPO?
- Can representations learned in a multimodal setting benefit vision-only deployment?
- Does attention across vision and touch lead to better overall performance?

### 6.1  COMPARED METHODS

We compare the following approaches:

- **M3L:** our approach jointly learns visual-tactile representations using a multimodal MAE and the policy using PPO.
- **M3L (vision policy):** while representations are trained from both visual and tactile data, the policy takes only visual data, exploiting the variable input length of the ViT encoder.
- **Sequential:** an M3L architecture trained independently for the different modalities in sequence. At each MAE training iteration, we first propagate the gradient for vision and then for touch. In this way, visual features cannot attend tactile features and vice versa.
- **Vision-only (w/ MAE):** an MAE approach with the same architecture as M3L, but trained only from visual inputs.
- **End-to-end:** a baseline that trains the policy end-to-end but with the same encoder architecture as M3L.

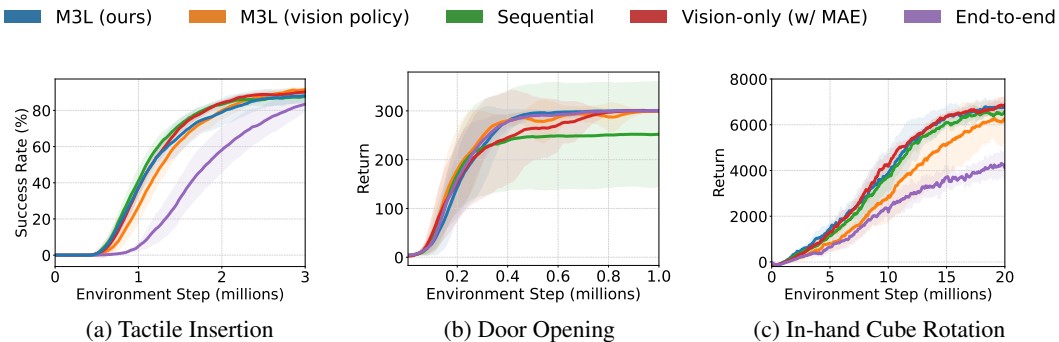

(a) Tactile Insertion      (b) Door Opening      (c) In-hand Cube Rotation

Figure 6: Learning curves investigating the advantages of M3L against baselines (see Section 6.1).

## 6.2 GENERALIZATION EXPERIMENTS

To evaluate the capabilities unlocked by multimodality, in this work we considered scenarios where both modalities are informative during most of the training episodes, i.e., visual information is most of the times sufficient to learn the task. Such a setting is especially suitable to isolate the effect of the multimodal representations (compared, for example, to the use of a single modality). In particular, we investigate the generalization capabilities unlocked by the multimodal representations when dealing with unseen objects or conditions. For the tactile insertion, we pretrain a policy on the set of 18 training objects, and test the zero-shot generalization on two different objects, which are a rectangular prism and V-shaped object (see Figure 4). Such objects are not seen during training, and the V-shaped object considerably differs from the training objects. For the door opening task, we randomize the initial position of the door, as well as the friction and damping coefficients of the hinges as described in Section 5.2. All of these parameters were instead fixed during training. Finally, for the in-hand rotation, we double the mass of the cube and slightly perturb the camera pose.

The results are shown in Figure 5, with M3L consistently competing with or outperforming the end-to-end baseline and all the other representation learning approaches on all tasks. In particular, M3L substantially outperforms the vision-only approach, exploiting the power of multimodal representations. While the sequential baselines is competitive with M3L on the tactile insertion and in-hand reorientation tasks, it performs considerably worse on the door opening task. In particular, sequential training largely degrades due to observed training instabilities (see Figure 6), indicating that attention across modalities enables the extraction of stronger and more general representations.

Interestingly, we observe a considerable improvement of M3L with vision policy over the vision-only baseline. They key insight is that using touch only for training the representation encoder is sufficient to substantially fill the gap with M3L on all tasks. This opens several remarkable opportunities, namely, I) a limited loss of generalization performance when touch is used at training time, but removed at deployment time, II) the possibility of training multimodal representations exclusively in simulation, and transferring a stronger vision policy to the real-world, wherever visual sim-to-real transfer is achievable (James et al., 2019).

## 6.3 TRAINING PERFORMANCE

We report the learning curves for each task in Figure 6. The methods based on representation learning typically exhibit higher sample efficiency compared to the end-to-end baseline, by exploiting the unsupervised reconstruction objective during training. M3L is the only approach that consistently achieves best in-task performance across the three tasks.

Finally, Figure 7 ablates the number of frames stacked together as an input to M3L (as explained in Section 4.1) for the tactile insertion task. The result may look counter-intuitive, given that the gripper is position controlled, and a single frame may appear sufficient to extract full information about the task. However,

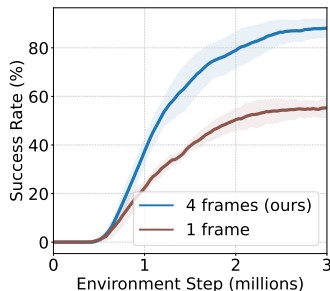

Figure 7: Frame stacking ablation

we hypothesize that when the framework is conditioned on a single frame, the encoder may struggle with visual occlusions. More importantly, contact information becomes much more relevant when stacking multiple frames, which act as a memory of a recent contact event. On the contrary, a single frame only signals current contact information, which immediately vanishes a step away from contact. Note that 4 frames are used as input to all the baselines considered in Figure 5 and Figure 6. Additional baselines and experiments are described in Appendix F and Appendix G.

## 7 CONCLUSION

We have presented a systematic representation learning approach, Masked Multimodal Learning (M3L), to fuse visual and tactile data when using reinforcement learning for manipulation tasks. The results indicate that in addition to being sample efficient compared to an end-to-end baseline, the multimodal representations improve generalization to unseen objects and conditions over a variety of baselines. We notably observed how the benefits of training multimodal representations is partly retained when the representation encoder is applied to a vision policy. Finally, while contributing to tasks that cannot be solved with vision alone is certainly an important application of tactile sensing, this work indicates that touch can considerably contribute to efficient and generalizable manipulation also for tasks where vision appears to be sufficient. Therefore, we hope that this work opens new perspectives to incorporate this modality in a wider range of applications and learning frameworks.

**Limitations and Future Work.** Our method suffers from some of PPO's drawbacks, e.g., higher sample complexity compared to off-policy algorithms and struggle with difficult exploration problems. However, the modularity of the representation learning block makes it possible to combine it with other RL algorithms, and this will be the subject of future work.

An additional limitation of our approach is that it uses tactile data at all times, even when such data are uninformative, e.g., when contact is not taking place, which can potentially lead to slowing down learning. This information sparsity has been investigated in the past and a plausible solution indicated as tactile gating (Hansen et al., 2022) may also be applied to our method.

Previous work that only relied on visual data (Xiao et al., 2022) leveraged MAEs in a pretraining fashion, with a large encoder trained off-domain and directly deployed for learning a variety of tasks. Part of this success is due to the large availability of image and video datasets available to the research community (Deng et al., 2009; Grauman et al., 2022). This is in contrast to the scarce availability of tactile datasets, often challenging to collect, especially when paired vision-touch data are required, such as for our approach. An interesting research direction would be to investigate how to leverage the large amount of available image data while only requiring a smaller portion of paired vision-touch data in a pretraining-finetuning fashion.

**Considerations for Real-World Application.** The current results were presented in simulation environments, which allowed us to thoroughly analyze and compare a wide range of architectural choices in a scalable manner. However, real-world applications may largely benefit from the findings of this work. Specifically, our algorithm shows improvements in sample efficiency compared to PPO from raw inputs (see Figure 6). Sample efficiency, together with the generalization properties showed by our approach, mark a crucial step towards the application of reinforcement learning on real-world robots, where we want to minimize both sample collection and retraining for each modification of the training task. Additionally, the performance benefits of using an M3L representation encoder for vision policies renders the possibility to train such policies in simulation with the availability of tactile signals, enabling the transfer of stronger vision policies to the real world, e.g., through the use of visual domain randomization.

Finally, the potential benefits of our work to real-world applications are confirmed by the successful transfer of approaches based on masked autoencoding from simulation to real-world systems in (Seo et al., 2023b; Radosavovic et al., 2023). In addition, the choice of force maps as tactile inputs has also proved its efficacy in sim-to-real transfer, as detailed in (Bi et al., 2021; Xu et al., 2023).

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

## A ALGORITHM

We list a detailed step-by-step overview of M3L in Algorithm 1.

---

**Algorithm 1** Masked Multimodal Learning (M3L)

---

1: Initialize MAE and PPO parameters
2: **repeat**
3:      // Collect rollouts
4:      Initialize rollout buffer $\mathcal{B} \leftarrow 0$
5:      **repeat**
6:          Read latest visual and tactile inputs
7:          Feed inputs through frozen networks without masking to compute representations $z$
8:          Use the current policy to compute the control action from $z$
9:          Store transition (with original inputs) to the rollout buffer
10:     **until** maximum of $N^{\text{PPO}}$ environment interactions
11:     // Update networks
12:     Train MAE and PPO using the latest rollouts for $M$ epochs following Figure 1
13: **until** maximum number $N^{\text{PPO}}_{\max}$ of environment interactions

---

## B BACKGROUND FOR PROXIMAL POLICY OPTIMIZATION

Let $\pi_\theta$ be a policy (or actor function) parameterized by $\theta$ that outputs an action $a_t$ given a state $s_t$, $V_{\theta_V}$ be a value (or critic) function parameterized by $\theta_V$, and $\hat{A}_t$ be be an estimator of the advantage function (Sutton & Barto, 2018) at a timestep $t$. The goal of Proximal Policy Optimization (PPO) (Schulman et al., 2017) is to address the problem of vanilla policy gradient update (Sutton & Barto, 2018) that often causes destructively large parameter updates. To this end, PPO introduces a new objective for training the actor that minimizes the following clipped surrogate objective, which is a lower bound of the conservative policy iteration objective (Kakade & Langford, 2002):

$$\mathcal{L}^{\text{clip}} = \hat{\mathbb{E}}_t \left[ \min(r_t(\theta)\hat{A}_t, \text{clip}(r_t(\theta), 1 - \epsilon, 1 + \epsilon)\hat{A}_t) \right] \tag{3}$$

where $\hat{\mathbb{E}}_t$ denotes an empirical average over a minibatch, $\epsilon$ is a hyperparameter, $r_t(\theta)$ is a probability ratio $\pi_\theta(a_t|s_t)/\pi_{\theta_{\text{old}}}(a_t|s_t)$, and $\theta_{\text{old}}$ are the actor parameters before the update. Note that this objective discourages large updates that would make the $r_t$ be outside the range of $(1 - \epsilon, 1 + \epsilon)$.

For training the critic function, PPO minimizes the following objective:

$$\mathcal{L}^{\text{critic}} = \hat{\mathbb{E}}_t \left[ \frac{1}{2}(V_{\theta_V}(s_t) - \hat{V}_t^{\text{targ}})^2 \right] \tag{4}$$

where $\hat{V}_t^{\text{targ}}$ is a target value estimate computed with the generalized advantage estimation (GAE) (Schulman et al., 2016), which is also used for computing $\hat{A}_t$. The final objective of PPO is given as follows:

$$\mathcal{L}^{\text{PPO}} = \mathcal{L}^{\text{clip}} + \beta_V \cdot \mathcal{L}^{\text{critic}} + \beta_H \cdot H[\pi] \tag{5}$$

where $H[\pi]$ is an action entropy bonus for exploration and $\beta_V, \beta_H$ are scale hyperparameters.

## C HYPERPARAMETERS

The training hyperparameters are listed in Table 1.

## D TACTILE INSERTION ENVIRONMENT

In Figure 8 we show all pegs and targets used for randomizing training in the tactile insertion environments. The dense reward used for the environment is the following:

$$r = -\log(100 \cdot d + 1) \tag{6}$$

where $d$ is the distance of the peg from the target.

Table 1: Hyperparameters

| Symbol | Description | Value |
|---|---|---|
|  | number of parallel PPO envs | 8 |
|  | masking ratio | 95% |
| $B$ | batch size | 512 |
| $n$ | number of representation learning steps for hand | 16 |
| $N^{\text{PPO}}$ | PPO rollout buffer length | 32768 for insertion and hand, 4800 for door |
| $M$ | PPO n. epochs | 10 |
|  | learning rate PPO | $10^{-4}$ |
| $\beta_T$ | tactile reconstruction weight | 10 |

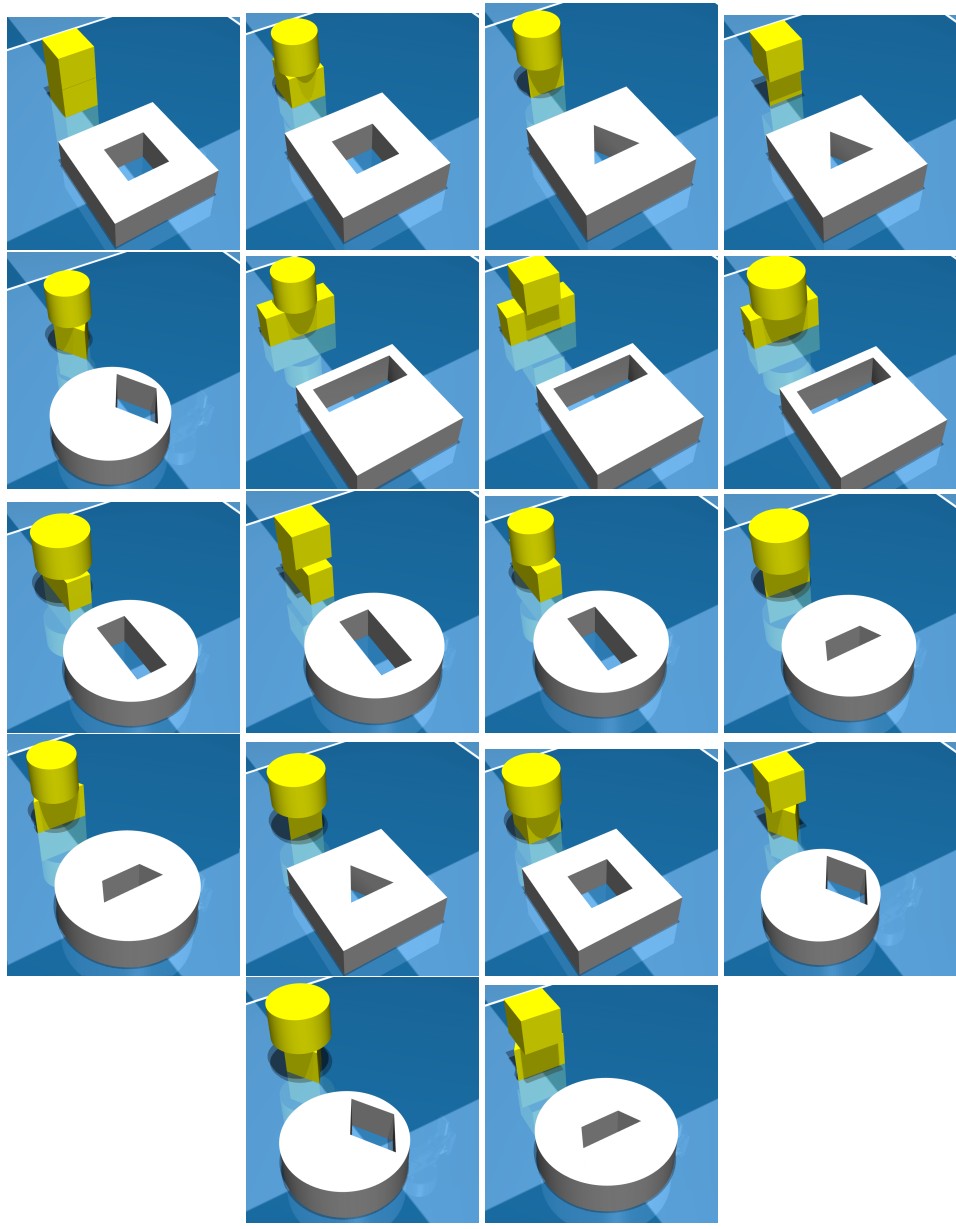

Figure 8: Objects (pegs) and targets used to train the tactile insertion task.

# E  IN-HAND CUBE ROTATION ENVIRONMENT

Figure 9: $32 \times 32$ tactile grid for the tactile observation in the in-hand cube rotation environment.

# F  ADDITIONAL BASELINES

This section presents additional baselines in addition to those described in Section 6.1. Note that experiments in this section have been executed on three random seeds.

In particular, we implemented a baseline that follows the concurrent work in Qi et al. (2023). In fact, we discretize the contact location, and feed a binarized representation to an MLP that create a tactile embedding of the stacked frames. Similarly, we feed the visual images to a CNN and concatenate the visual embedding with the tactile embedding. We feed the concatenation into a similar transformer encoder that outputs representations that we input to PPO. This baseline is then trained end-to-end on the insertion task (without employing auxiliary objectives or privileged learning). The results are shown in Figure 10a, with such a baseline underperforming both M3L and the vision-only baseline co-trained with an MAE objective.

Moreover, we compare M3L with two vision-only baselines that rely on frozen pretrained representations, specifically using the visual encoders from Masked Visual Pretraining (MVP, Xiao et al. (2022); Radosavovic et al. (2023)) and Contrastive Language-Image Pretraining (CLIP, Radford et al. (2021)). Note that both encoders are much larger than the one used in M3L (5M for M3L, 22M for MVP, and 88M for CLIP), leading to considerably slower training. The results are shown in Figure 10b, with both M3L and its vision-only MAE variation outperforming MVP and CLIP representation on the in-domain learning task.

Finally, we compare M3L with a touch-only MAE-based baseline (similar to vision-only w/ MAE) on the tactile insertion and door opening tasks. Note that the insertion and task is misspecified when vision is missing, because of missing target information, while the door task is not necessarily requiring vision to be solved. Results are shown in Figure 11, with such a baseline underperforming M3L, particularly on the insertion task.

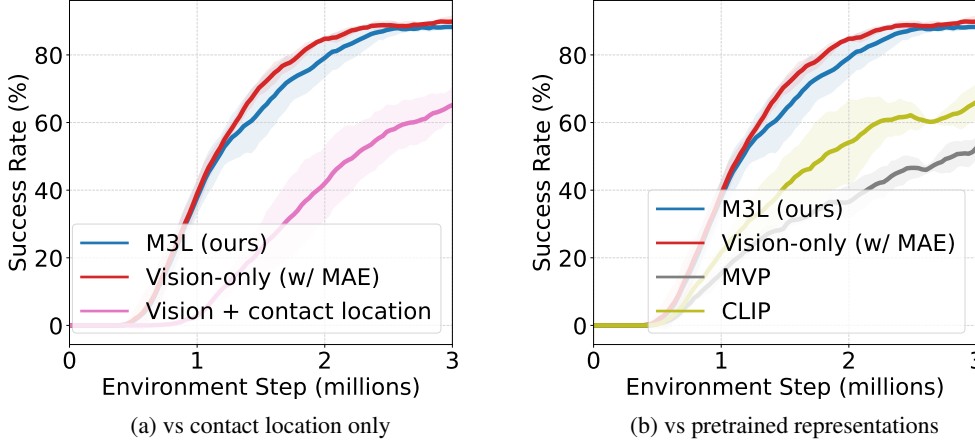

Figure 10: In-domain (tactile insertion) learning curves comparing M3L with baselines.

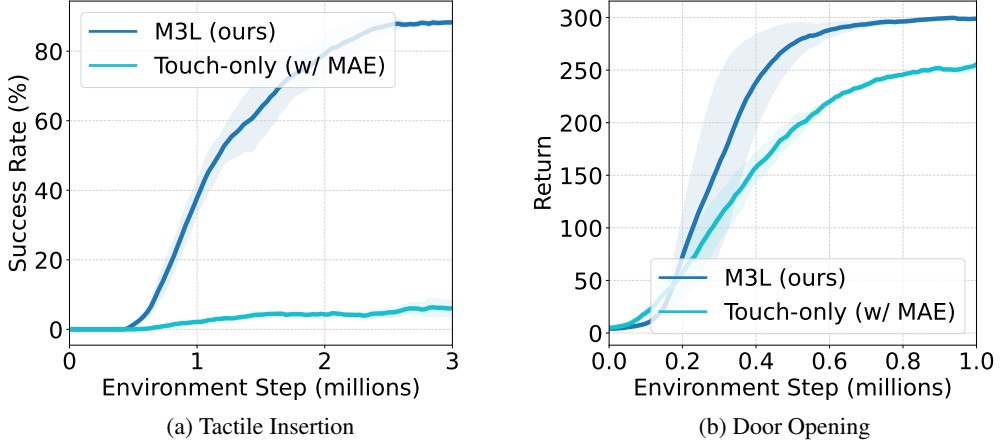

Figure 11: In-domain learning curves comparing M3L with touch-only baseline.

## G   ROBUSTNESS TO NOISE

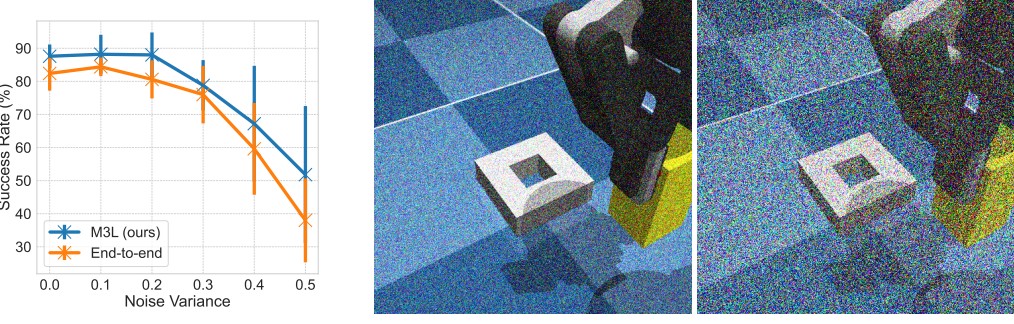

(a) Performance vs noise        (b) Image with 0.2 noise variance (c) Image with 0.5 noise variance

Figure 12: Noise robustness analysis.

We test the robustness of the trained policies by adding zero-mean Gaussian noise to both visual and tactile data at inference time. Figure 12a shows the performance of M3L vs the end-to-end for increasing noise variance, highlighting how M3L success rates do not degrade up to a reasonable degree of disturbances and also consistently outperforms the end-to-end approach also in term of robustness. Note from Figure 12b and Figure 12c how performance only degrades for considerable perturbations of the original image and tactile data.

## H   MAE RECONSTRUCTIONS

Example reconstruction of both visual and tactile inputs are shown in Figure 13.

## I   ADDITIONAL TASKS

In this section, we present two additional tasks, **Egg Rotation** and **Pen Rotation**. These tasks are essentially performed in the same environment as the in-hand rotation task described in Section 5.3. However, we replace the cube with an egg and a pen, respectively (see Figure 14). In these new tasks, we test *visual variations*, e.g., the camera pose in the egg task and the color of the pen in the pen task.

The results are shown in Figure 15. M3L outperforms the baselines in these environments in terms of generalization. In the egg rotation task, where we perturbed the camera pose, M3L vision policy outperforms the vision-only (w/ MAE) approach during training as well as testing. Interestingly, touch is even more crucial for generalization in the pen task, where the pen has an unseen color, largely outperforming M3L vision policy. We believe this is due to the fact that the multimodal policy learns to rely less on vision (having the possibility of using touch too), which leads to more visual robustness even compared to M3L vision-policy. As a sanity check for this, we also tested feeding a constant visual input (i.e. an initial image) to M3L, which led to a major drop in performance (below all other baselines and not shown in the figure). This confirms that M3L effectively leverages both vision and touch to learn robust policies.

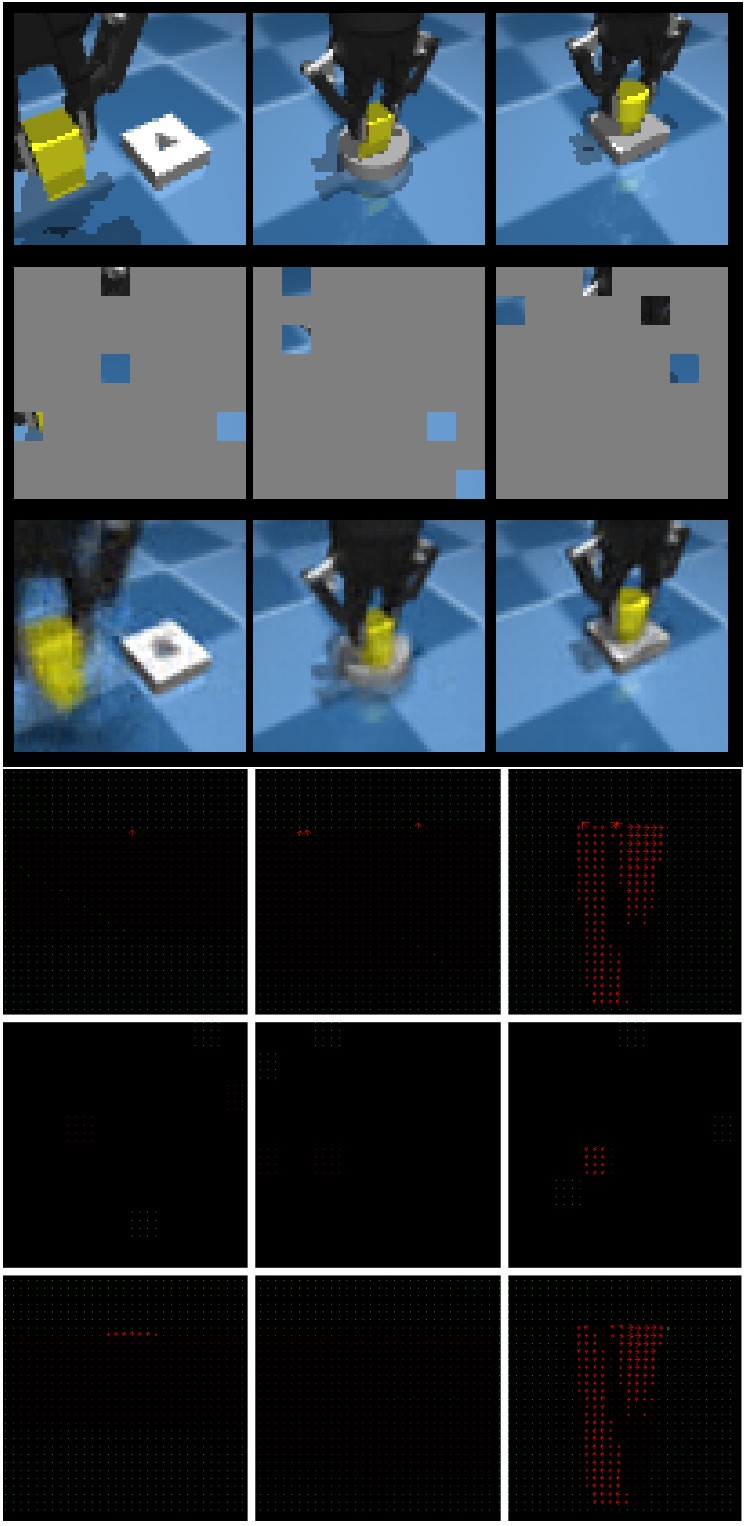

Figure 13: Examples of MAE reconstructions for visual and tactile inputs. First and fourth rows are original inputs. Second and fifth rows are visualizations of the masked and unmasked features (note that we mask convolutional features and not raw input). Third and sixth rows are reconstructions.

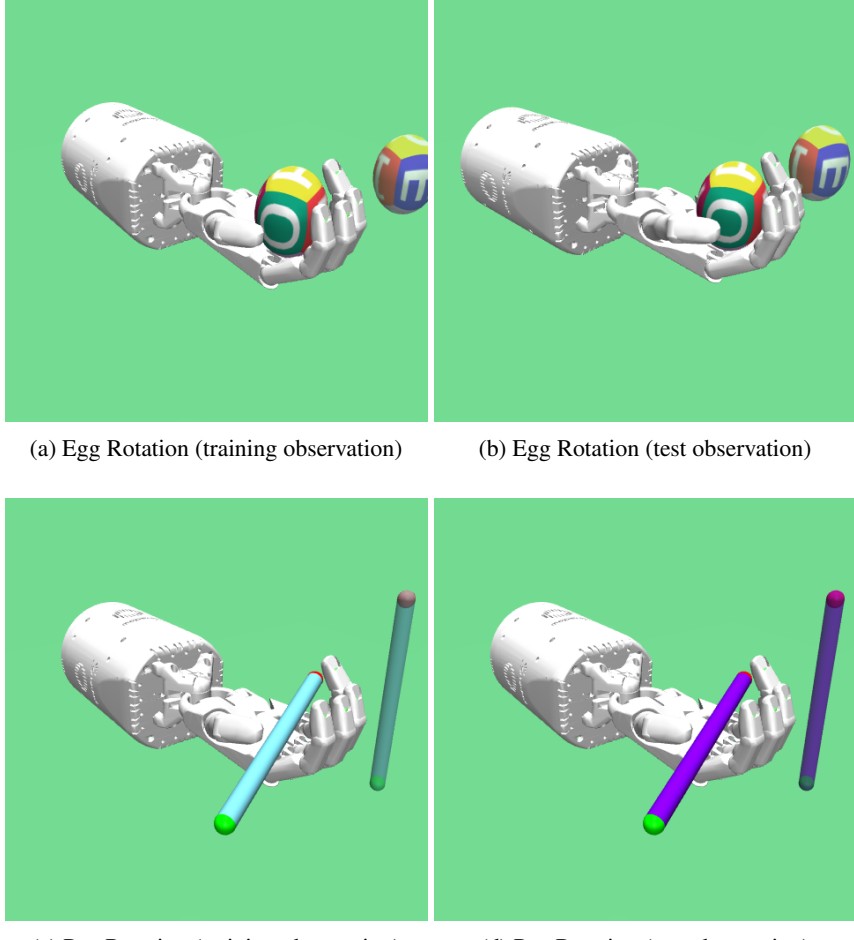

(a) Egg Rotation (training observation)      (b) Egg Rotation (test observation)

(c) Pen Rotation (training observation)      (d) Pen Rotation (test observation)

Figure 14: Snapshots from the in-hand egg and pen rotation environments.

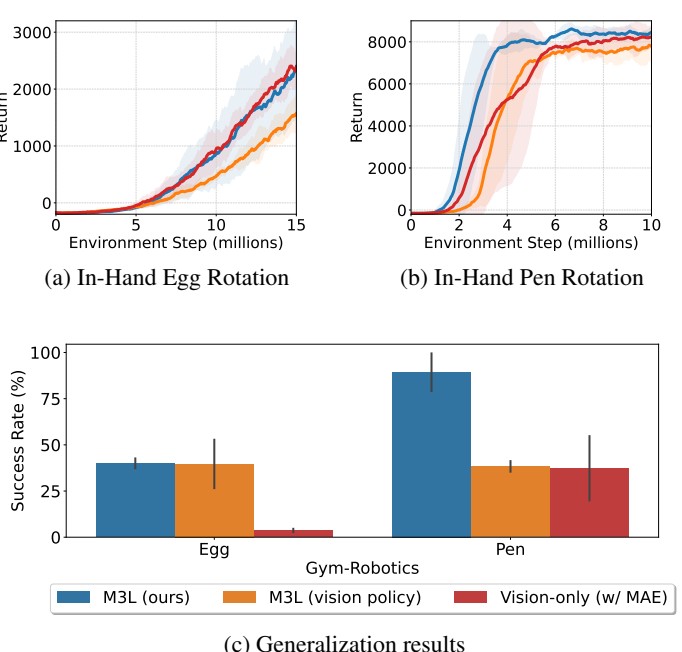

(a) In-Hand Egg Rotation

(b) In-Hand Pen Rotation

(c) Generalization results

Figure 15: In-domain (first row) and generalization (second row) performance on the in-hand egg and pen rotation tasks.

