# OpenReview forum: "The Power of the Senses: Generalizable Manipulation from Vision and Touch through Masked Multimodal Learning"
_ICLR.cc/2024/Conference — Submitted to ICLR 2024_

### Official Review · Reviewer_HDvH · 2023-10-29

**Soundness:** 3 good
**Presentation:** 3 good
**Contribution:** 1 poor
**Rating:** 3
**Confidence:** 3

**Summary:**

The authors introduced a technique that merges multimodal self-supervised representation learning with robot policy learning through the PPO algorithm. This approach showcases that by utilizing masked autoencoding for both policy and visual-tactile representations, one can enhance sample efficiency and achieve superior generalization capabilities.

**Strengths:**

1. The content is articulately written, making it easily comprehensible.
2. The appendix offers in-depth details about both the model and the environment, ensuring reproducibility of the work.
3. The model innovatively incorporates multimodal representation features to address robot learning tasks, marking a significant stride in this research direction.

**Weaknesses:**

1. A prevalent challenge in contemporary research within this domain is tactile simulation. The tactile simulation employed in this study lacks the desired accuracy. Moreover, the introduced method hasn't been empirically tested or combined with tactile sensors in real-world experiments, which is a missing link for comprehensive validation.
2. The ViT model, as utilized, is computationally intensive and has a propensity to overfit to specific problems. The potential benefits of ViT's self-supervised pre-training techniques, such as MAE, ought to be realized across a diverse range of datasets or environments. Given the constraints observed in the study—limited tasks, a restricted set of objects, and minimal variations in background textures—it's challenging to fully endorse the promise of the proposed approach.
3. As discerned from Figure 6, there's an inconsistency in the results between the baseline and the proposed method across various tasks. This discrepancy muddles the clarity needed to conclusively determine the efficacy of the proposed method. It would be advisable to extend the scope by implementing over 10 distinct tasks and undertaking comprehensive statistical analyses to more convincingly demonstrate the method's effectiveness.

**Questions:**

Please see the weaknesses above.

---

> ### Author Response · Authors · 2023-11-18
> **Response**
>
> **#1: The tactile simulation employed in this study lacks the desired accuracy. Moreover, the introduced method hasn't been empirically tested or combined with tactile sensors in real-world experiments.**
>
> MuJoCo’s contact force optimization is equivalent to a penalty-based approach, which is essentially the same used to compute force maps in [A] and [B], with those works showing zero-shot transfer to the real-world for the specific tasks considered. Additionally, there are other remarkable examples of sim-to-real transfer using MuJoCo [C], [D]. Based on such evidence, we would like to highlight that MuJoCo’s touch grid is currently the most realistic general-purpose simulator of tactile maps, especially given the wide range of possibilities for proper system identification. We expect this very important clarification to change the reviewer’s evaluation of the contribution.
>
> While the main focus of our work was on studying how to extract stronger representations when visual and tactile data are available, we also showed some promising direction (e.g., improvement of vision-only policies with tactile-enabled representation training) for real-world deployment, which we leave as future work.
>
> **#2: The ViT model, as utilized, is computationally intensive and has a propensity to overfit to specific problems. The potential benefits of ViT's self-supervised pre-training techniques, such as MAE, ought to be realized across a diverse range of datasets or environments.**
>
> Thanks for suggesting this comparison! We indeed compared to MVP [E], which is basically relying on ViT representations pretrained across datasets, and on an additional baseline using CLIP [F] visual encoder. As shown in Figure 10b of the updated appendix, both M3L and its vision-only MAE variation outperform MVP and CLIP representations on the in-domain learning task (tactile insertion). We believe this is due to the fact that we let the representations further adapt during task learning. Note that training such MVP and CLIP baselines was even more computationally intensive than training our all-in-one approach, due to the inference of the much larger MVP (22M) and CLIP (88M) encoders, compared to the small ViT used in M3L (5M).
>
> We omit generalization results on the new baselines due to the low success rates on the in-domain task.
>
> **#3: There's an inconsistency in the results between the baseline and the proposed method across various tasks. It would be advisable to extend the scope by implementing over 10 distinct tasks and undertaking comprehensive statistical analyses to more convincingly demonstrate the method's effectiveness.**
>
> Our results show that 1) vision+touch policies perform best when investigating zero-shot generalization, 2) training representations with vision+touch also improve vision policies, 3) representation learning approaches outperform end-to-end approaches when learning in-domain. 1) and 2) are shown in Figure 5 in the paper, while 3) is shown in Figure 6 in the paper. These results are consistent across all tasks, so we disagree with the reviewer comment.
>
> As the reviewer pointed out in #1, the use of tactile simulation is still at an early stage. While we agree that evaluation on a large variety of tasks would be meaningful, note that our three environments are the first available implementations of RL environments with availability of both visual and high-resolution tactile sensing on a general-purpose like MuJoCo. This required carefully designing and adapting each of the tasks for this work, as opposed to other fields in RL that have the possibility of benchmarking on existing environments. To give an idea of the process involved, to increase the number of contact points on the Shadow Hand, we needed to split each collision mesh in a much higher number of meshes making sure that each of these was still convex (as required by MuJoCo), which is a non-trivial process for each robot embodiment. Ours is an effort in this direction, and open-sourcing such environments will hopefully give the opportunity to fellow researchers to further extend the benchmarking opportunities in the area. As the reviewer pointed out, however, we purposely focused on tasks that are very different to cover a wider spectrum of applications. We are currently working on further extending the number of tasks.

---

> > ### Author Response · Authors · 2023-11-18
> > **Response (references)**
> >
> > [A] Bi, Thomas, et al. "Zero-shot sim-to-real transfer of tactile control policies for aggressive swing-up manipulation." IEEE Robotics and Automation Letters 6.3 (2021): 5761-5768.
> >
> > [B] Xu, Jie, et al. "Efficient tactile simulation with differentiability for robotic manipulation." Conference on Robot Learning. PMLR, 2023.
> >
> > [C] Akkaya, Ilge, et al. "Solving rubik's cube with a robot hand." arXiv preprint arXiv:1910.07113 (2019).
> >
> > [D] Haarnoja, Tuomas, et al. "Learning agile soccer skills for a bipedal robot with deep reinforcement learning." arXiv preprint arXiv:2304.13653 (2023).
> >
> > [E] Radosavovic, Ilija, et al. "Real-world robot learning with masked visual pre-training." Conference on Robot Learning. PMLR, 2023.
> >
> > [F] Radford, Alec, et al. "Learning transferable visual models from natural language supervision." International conference on machine learning. PMLR, 2021.

---

> ### Author Response · Authors · 2023-11-22
> **Additional tasks**
>
> **Additional tasks**
>
> We performed additional experiments on two new in-hand rotation tasks, namely, egg and pen rotation tasks.
>
> These tasks confirm that M3L outperforms the baselines in terms of generalization, particularly by a large margin when altering the pen color in the corresponding rotation task. Please find the corresponding discussion in Appendix I of the paper. We hope that this clarifies your related concern. We look forward to hearing your feedback before the end of the rebuttal period.

---

### Official Review · Reviewer_pVq9 · 2023-10-31

**Soundness:** 2 fair
**Presentation:** 3 good
**Contribution:** 3 good
**Rating:** 5
**Confidence:** 4

**Summary:**

This paper proposes an approach for combining visual and tactile data in training robot manipulation policies in simulation. The main contribution of the paper is a representation learning algorithm (M3L) based on masked auto-encoding that can ingest both image observations and tactile observations (taxels). The policy learning with the learned representations can be performed with existing reinforcement learning algorithms like PPO. Experiments on three robot simulation tasks show that the proposed approach is effective in distilling multimodal touch+vision representations for policy learning.

**Strengths:**

- The paper presents a neat idea of combining touch and vision through an existing paradigm of masked autoencoding, and showing that this is helpful for simulated robot manipulation tasks. The overall idea is simple and can be used widely with minimal modifications.

- The simulation results are interesting in terms of the same algorithm being able to learn policies for very different tasks when trained separately on each simulation environment. It is good to note that these tasks span fine-grained manipulation, articulated object opening, and dexterous manipulation.

- The overall paper is well-written and insights are communicated clearly. Sufficient details are provided in the Appendix, and the video results are helpful in understanding the tasks.

**Weaknesses:**

1. The paper ignores several different related works, both in terms of tactile sensors used for robotics, and machine learning approaches that use touch as a modality for manipulation tasks. Here are some missing papers that should be discussed:

         a) robot learning papers with tactile data [1,2] these papers show real-robot dexterous manipulation tasks with either tactile data along or with both vision and tactile data (followed by RL with sim2real transfer)
         b) versatile tactile sensors for real-world manipulation [3]

2. The simulation results are interesting in terms of the same algorithm being able to learn policies for very different tasks (as mentioned in the strengths) but it is unclear to me why experiment with only three tasks? Most reinforcement learning in simulation papers show results on a large number of environments, and this is important to understand the capabilities of the proposed approach, without over-indexing on a particular environment.

3. There are no external baselines compared against. All the comparisons are with variants of the proposed M3L algorithm. The related works in representation learning that use vision-only observations, including papers that do pre-training for representation learning are relevant for empirical comparisons (for example [4,5])

4. The introduction mentions "M3L demonstrates better zero-shot generalization to unseen objects and variations of
the task scene" however the experiments show only very weak generalization to minor changes in the object (in particular mass, damping, friction variations). Hence the claim is not really substantiated in my understanding. It is important to show generalization to actually different objects (for example with geometric variations like shape and size, in addition to the current variations). Also, the results should evaluate this for a number of different objects and not just a single new object, in order for the conclusions to be statistically significant.


[1] Qi, Haozhi, Brent Yi, Sudharshan Suresh, Mike Lambeta, Yi Ma, Roberto Calandra, and Jitendra Malik. "General in-hand object rotation with vision and touch." arXiv preprint arXiv:2309.09979 (2023).

[2] Yin, Zhao-Heng, Binghao Huang, Yuzhe Qin, Qifeng Chen, and Xiaolong Wang. "Rotating without Seeing: Towards In-hand Dexterity through Touch." arXiv preprint arXiv:2303.10880 (2023).

[3] Bhirangi, Raunaq, Tess Hellebrekers, Carmel Majidi, and Abhinav Gupta. "ReSkin: versatile, replaceable, lasting tactile skins." arXiv preprint arXiv:2111.00071 (2021).

[4] Nair, Suraj, Aravind Rajeswaran, Vikash Kumar, Chelsea Finn, and Abhinav Gupta. "R3m: A universal visual representation for robot manipulation." arXiv preprint arXiv:2203.12601 (2022).

[5] Radosavovic, Ilija, Tete Xiao, Stephen James, Pieter Abbeel, Jitendra Malik, and Trevor Darrell. "Real-world robot learning with masked visual pre-training." In Conference on Robot Learning, pp. 416-426. PMLR, 2023.

**Questions:**

Refer to the list of weakness for more details.

1. Is it possible to provide a more detailed treatment of related works? In particular, robot learning papers with tactile data [1,2] and versatile tactile sensors for real-world manipulation [3] are relevant, and missing.

2. Is it possible to have more than three simulation tasks? Most reinforcement learning in simulation papers show results on a large number of environments, and this is important to understand the capabilities of the proposed approach, without over-indexing on a few particular environments.

3. Is it possible to compare with external baselines? The related works in representation learning that use vision-only observations, including papers that do pre-training for representation learning are relevant for empirical comparisons (for example [4,5])

4. Is it possible to evaluate generalization with different objects (shapes, sizes, visual appearances) ? Since generalization is a main claim of the paper, this seems particularly important.

5. Real-World tactile data is usually very noisy, but simulation data is clean. Is there anything that will need to be changed in the approach for bridging this gap or making the representations more robust to noise, when real-world deployment is attempted? In general, the results in the paper are not convincing for justifying the last paragraph of page 9, because there is very little generalization demonstrated in simulation itself.

---

> ### Author Response · Authors · 2023-11-18
> **Response**
>
> **#1: The paper ignores several different related works [A], [B], [C], both in terms of tactile sensors used for robotics, and machine learning approaches that use touch as a modality for manipulation tasks.**
>
> Thank you for pointing out relevant works. We have included these works in our updated related work section and also compared our method with [A] in Figure 10, Appendix. However, please note that [A] is a contemporaneous work according to ICLR guidelines, having been arxived only 10 days prior to the submission of this paper.
>
> **#2: Is it possible to have more than three simulation tasks?**
>
> We agree that evaluation on a large variety of tasks would be meaningful and thus, we are currently working on further extending the number of tasks.
>
> We want to emphasize that our three environments are the **first** RL environments both with visual and **high-resolution tactile** sensing on a general-purpose physics engine like MuJoCo. This required carefully designing and adapting each of the tasks for this work, as opposed to other fields in RL that have the possibility of benchmarking on existing environments. To give an idea of the process involved, to increase the number of contact points on the Shadow Hand, we needed to split each collision mesh in a much higher number of meshes making sure that each of these was still convex (as required by MuJoCo), which is a non-trivial process for each robot embodiment. Ours is an effort in this direction, and open-sourcing such environments will hopefully give the opportunity to fellow researchers to further extend the benchmarking opportunities in the area. As the reviewer pointed out, however, we purposely focused on tasks that are very different amon each other to cover a wider spectrum of applications.
>
> **#3: There are no external baselines compared against. All the comparisons are with variants of the proposed M3L algorithm.**
>
> As suggested by the reviewer, we have included comparisons to other approaches in Figure 10 of the updated version: contact location-based tactile representations [A] and vision-only pre-trained representations (MVP [D], CLIP [E]).
>
> While [A] is contemporaneous work, we found it interesting to compare with a similar transformer architecture, trained in a single stage from vision and touch. In particular, we also investigate the tactile representation used in [A], that is, discretized contact locations. In contrast, our approach uses intensity and directions of spatially distributed force vectors (force maps). The results are shown in Figure 10a of the updated appendix, with the discretized contact location approach outperformed by both our M3L and end-to-end baselines even for the in-domain learning task (tactile insertion).
>
> We also added comparisons to baselines based on vision-only representations extracted through MVP [D] and CLIP [E] visual encoders, kept frozen during PPO training. Note that both encoders are much larger than the one used in M3L (5M for M3L, 22M for MVP, and 88M for CLIP), leading to considerably slower training. The results are shown in Figure 10b of the updated appendix, with both M3L and its vision-only MAE variation outperforming MVP and CLIP representation on the in-domain learning task (tactile insertion).
>
> We omit generalization results on the new baselines due to the low success rates on the in-domain task.
>
> **#4: The experiments show only very weak generalization to minor changes in the object (in particular mass, damping, friction variations). It is important to show generalization to actually different objects.**
>
> In the tactile insertion task, we indeed test the generalization capability to **different objects**, as illustrated in Figure 4 and Figure 8.
>
> Moreover, we would argue that other perturbations in the objects are not minor, i.e., 10x friction and damping, 2x weight. In addition, we test on different object poses in Door Opening and camera poses in In-Hand Rotation.
>
> **#5: Real-World tactile data is usually very noisy, but simulation data is clean. Is there anything that will need to be changed in the approach for bridging this gap or making the representations more robust to noise, when real-world deployment is attempted?**
>
> Thanks for the great suggestion! We added a noise robustness analysis in Figure 12 of the updated appendix, where we add zero-mean Gaussian noise to both visual and tactile data at inference time. The figure shows the performance of M3L vs the end-to-end for increasing noise variance, highlighting how M3L success rates do not degrade up to a reasonable degree of disturbances and also consistently outperforms the end-to-end approach also in terms of robustness.
>
> Also, note that force maps have already been shown to be transferable to the real-world, either by estimating them [F] or by mapping them to optical flow [G].

---

> > ### Author Response · Authors · 2023-11-18
> > **Response (references)**
> >
> > [A] Qi, Haozhi, et al. "General in-hand object rotation with vision and touch." arXiv preprint arXiv:2309.09979 (2023).
> >
> > [B] Yin, Zhao-Heng, et al. "Rotating without Seeing: Towards In-hand Dexterity through Touch." arXiv preprint arXiv:2303.10880 (2023).
> >
> > [C] Bhirangi, Raunaq, et al. "ReSkin: versatile, replaceable, lasting tactile skins." arXiv preprint arXiv:2111.00071 (2021).
> >
> > [D] Radosavovic, Ilija, et al. "Real-world robot learning with masked visual pre-training." Conference on Robot Learning. PMLR, 2023.
> >
> > [E] Radford, Alec, et al. "Learning transferable visual models from natural language supervision." International conference on machine learning. PMLR, 2021.
> >
> > [F] Bi, Thomas, et al. "Zero-shot sim-to-real transfer of tactile control policies for aggressive swing-up manipulation." IEEE Robotics and Automation Letters 6.3 (2021): 5761-5768.
> >
> > [G] Xu, Jie, et al. "Efficient tactile simulation with differentiability for robotic manipulation." Conference on Robot Learning. PMLR, 2023.

---

> > ### Comment · Reviewer_pVq9 · 2023-11-23
> > **response**
> >
> > Dear authors, thank you for your rebuttal responses and update to the paper. The additional comparisons with vision-only pre-training are helpful however several of my other concerns have not been adequately addressed. In particular, I am not convinced regarding the generalization to different objects, and real world applicability. Similar concerns about real world deployment have also been raised by reviewer bnCo which I agree with. Hence, I am unable to recommend accepting the paper.

---

> ### Author Response · Authors · 2023-11-22
> **Additional tasks**
>
> **Additional tasks**
>
> We performed additional experiments on two new in-hand rotation tasks, namely, egg and pen rotation tasks.
>
> These tasks confirm that M3L outperforms the baselines in terms of generalization, particularly by a large margin when altering the pen color in the corresponding rotation task. Please find the corresponding discussion in Appendix I of the paper. We hope that this clarifies your related concern. We look forward to hearing your feedback before the end of the rebuttal period.

---

### Official Review · Reviewer_bnCo · 2023-11-01

**Soundness:** 1 poor
**Presentation:** 2 fair
**Contribution:** 2 fair
**Rating:** 5
**Confidence:** 4

**Summary:**

This work proposes to learn visuo-tactile representation for manipulation with mask autoencoding. By utilizing MAE objective and structure, the learned joint representation improves the performance, sample efficiency and generalization ability of the policy. The method is evaluated in simulation for tactile insertion, door opening and in-hand cuda rotation tasks. Results also show the joint representation improves the vision-only policy performance in the test time.

**Strengths:**

1. Utilizing MAE architecture for modeling multimodal information for manipulation is intuitive and easy to understand. MAE objective for modeling joint representations is also a natural choice.
2. This work provides enough technical details, hyper-parameters, and visualization for reproducing the results.
3. This provides an interesting result that with multimodal self-supervised learning, the performance of vision-only policy improved during the test.

**Weaknesses:**

1. Manipulation with tactile sensor and visual input has been evaluated in the real world for different tasks[1]. To actually validate the performance of the proposed method, the real world evaluation is required. Especially for noisy tactile sensory information.
2. To validate the effectiveness of MAE architecture and self-supervised objective in this application, additional baselines (not limited to the baselines i mentioned) need to be compared. Qi, et al utilize a visuotactile transformer to model sequential observations. Hansen, et al show promising result in applying self-supervised learning in visual RL.
3. Missing tactile-only baseline, to fully address the essence of the multimodal representation.
4. Additional arm (or hand) proprioception information should be included in this kind of multi-modal manipulation pipeline, since it’s much easier to get (in sim and real) and provide rich information for manipulation tasks.
5. Out of three tasks, the proposed framework only outperforms vision-only baselines in In-hand rotation tasks. It seems tactile insertion and door-opening tasks are not quite tasks to evaluate the additional usage of tactile information.
6. For generalization experiments, randomization on on physical parameters might not be enough. Adding different level of noise to the visual observation and tactile reading and test the generalization on that end could further validate the effectiveness of the proposed self-supervised pipeline.


Reference:

[1] Qi, et al. General In-Hand Object Rotation with Vision and Touch

[2] Hansen, et al. Stabilizing Deep Q-Learning with ConvNets and Vision Transformers under Data Augmentation

**Questions:**

1. It would be great for authors to address the things mentioned in the weakness section
2. The result in FIgure 10, shows the learned MAE could reconstruct almost the full observation with quite limited observation. I’m wondering whether the learned model can show the same level of performance if there are noise in visual observation or tactile reading. Otherwise it looks lille the model is overfitting to the training distribution.

---

> ### Author Response · Authors · 2023-11-18
> **Response**
>
> **#1: Manipulation with tactile sensor and visual input has been evaluated in the real world for different tasks [A]. To actually validate the performance of the proposed method, the real world evaluation is required. Especially for noisy tactile sensory information.**
>
> We would like to highlight that [A] is a contemporaneous work according to ICLR guidelines, having been arxived only 10 days prior to the submission of this paper. We are however happy to cite and compare to [A] (see also point #2).
>
> [A] trained an end-to-end baseline (in two privileged learning stages) and achieved sim-to-real transfer through large-scale GPU parallelization and domain randomization. The focus of our work was instead on studying how to extract stronger representations when visual and tactile data are available. We also showed some promising direction (e.g., improvement of vision-only policies with tactile-enabled representation training) for real-world deployment, which we leave as future work.
>
> Regarding the robustness to noise, please see our response to #7.
>
> **#2: Additional baselines need to be compared.**
>
> Thank you for your suggestion. We have compared our method with additional baselines: contact location-based tactile representations [A] and vision-only pre-trained representations (MVP [B], CLIP [C]).
>
> Figure 10a of the updated appendix shows that both our M3L and end-to-end baselines outperform the contemporaneous work that uses discretized contact location approach [A] for the in-domain learning task (tactile insertion), confirming the advantages of using high-resolution force maps as opposed to low dimensional contact quantities.
>
> The pre-trained representations, MVP [B] and CLIP [C], also show slower training curves in Figure 10b. Note that the encoders used in MVP (22M parameters) and CLIP (88M parameters) are much larger than our M3L (5M parameters).
>
> We omit generalization results on the new baselines due to the low success rates on the in-domain task.
>
> **#3: Missing tactile-only baseline.**
>
> This is a great point! We have run additional experiments with the tactile-only baseline. Interestingly, although the Door Opening task can be fully solved without vision, M3L (vision + tactile) achieves much faster and better learning results than the tactile-only baseline. The benefit of M3L could come from (1) our multimodal representation learning and (2) the use of visual observations. We included this result in Figure 11 of the updated paper.
>
> Note that the insertion and hand tasks are misspecified when vision is missing, because of missing target and incomplete state information. We ran such a baseline on the insertion task to confirm such a claim (see Figure 11).
>
> **#4: Adding proprioception information.**
>
> We agree that proprioception (and any other sensing modalities) can certainly help robotic manipulation. However, this study aimed to isolate the contributions of vision and touch for representation learning, which is why we focus on these two modalities. We leave integration of proprioception and other modalities into our pipeline as future work.
>
> **#5: Out of three tasks, the proposed framework only outperforms vision-only baselines in In-hand rotation tasks.**
>
> Our results show that 1) vision+touch policies perform best when investigating zero-shot generalization, 2) training representations with vision+touch also improve vision policies, 3) representation learning approaches outperform end-to-end approaches when learning in-domain. 1) and 2) are shown in Figure 5 in the paper, while 3) is shown in Figure 6 in the paper. These results are consistent across all tasks, so we disagree with the reviewer comment. Please let us know if you have any other concerns regarding this.
>
> **#6: For generalization experiments, randomization on physical parameters might not be enough.**
>
> We do not only have randomization on physical parameters, but also variation of object pose (door) and camera pose (hand) in the generalization experiments. On all such randomizations, M3L shows the smallest performance degradation compared to baselines.
>
> **#7: Can the learned model show the same level of performance if there is noise in visual observation or tactile reading?**
>
> Thanks for the great suggestion! We added a noise robustness analysis in Figure 12 of the update appendix, where we add zero-mean Gaussian noise to both visual and tactile data at inference time. The figure shows the performance of M3L vs the end-to-end for increasing noise variance, highlighting how M3L success rates do not degrade up to a reasonable degree of disturbances and also how M3L consistently outperforms the end-to-end approach in terms of robustness.

---

> > ### Author Response · Authors · 2023-11-18
> > **Response (references)**
> >
> > [A] Qi, Haozhi, et al. "General in-hand object rotation with vision and touch." arXiv preprint arXiv:2309.09979 (2023).
> >
> > [B] Radosavovic, Ilija, et al. "Real-world robot learning with masked visual pre-training." Conference on Robot Learning. PMLR, 2023.
> >
> > [C] Radford, Alec, et al. "Learning transferable visual models from natural language supervision." International conference on machine learning. PMLR, 2021.

---

> > > ### Comment · Reviewer_bnCo · 2023-11-21
> > > **Response**
> > >
> > > I appreciate authors for addressing many of my concern regarding this work, including additional baseline, visualization.
> > >
> > > However, for a work regarding tactile sensing, It's necessary to show results in the real world (with proprioception). for the following reasons:
> > > 1. For most of robotics task, (if not all), proprioception of the robot is accessible and much easier to get compared to tactile and visual information. If proposed method only use vision or tactile information, it's possible the performance gain is from recovering the information loss for not using proprioception information. Overall, it's hard to evaluate the contribution of the work without proprioception in robotics work.
> > > 2. Tactile sensing is known to be noisy and hard to transfer to the real world. Since this work is addressing the usage of tactile sensing for manipulation, it's necessary to show it in the real world. Simulation-only results can hardly show whether it's actually working or not.
> > >
> > > In this case, I'll keep my original evaluation.

---

> > > > ### Author Response · Authors · 2023-11-22
> > > > **Additional clarification**
> > > >
> > > > Thank you for your response. We are glad that our previous response covered most of your concerns.
> > > >
> > > > Regarding the remaining concern on transferring tactile readings to the real world, in addition to the noise robustness analysis that you suggested, we would like to note that one of the findings of our work is that our approach also improves vision-only policies for certain tasks. In other words, even using simulated touch has the potential to improve vision policies, whose transfer to the real world has been extensively studied in the literature.

---

### Author Response · Authors · 2023-11-18
**Rebuttal**

We would like to thank all reviewers for their in-depth analysis and helpful comments. In addition to the point-to-point responses, we have increased the statistical significance of our experiments by expanding on the number of random seeds to 5 and training the hand experiment for 20M environment steps.

We have also updated our paper and highlighted the changes in yellow:
* Updated results with 5 random seeds in Figure 5, 6, and 7
* Added four additional baselines in Appendix F
* Added noise robustness experiment in Appendix G
* Additional references

 We are looking forward to an engaging discussion.

---

### Meta-Review · Area_Chair_76LU · 2023-12-03

**Metareview:**

The paper received one Reject and two Marginally Below Acceptance ratings. Reviewer bnCo expressed concerns regarding real-world evaluation, missing baselines, and generalization experiments. Reviewer pVq9 pointed out issues with the incomplete review of related work, a limited number of tasks, and missing baselines. Additionally, Reviewer HDvH highlighted weaknesses such as the absence of real-world experiments, a limited range of tasks, a restricted set of objects, and inconsistencies in the results. Despite the authors providing a rebuttal that addressed some concerns, the reviewers remained unconvinced. The AC concurs with the weaknesses identified by the reviewers, leading to the recommendation for rejection.

**Justification For Why Not Higher Score:**

The paper in its current form has various issues which preclude acceptance.

**Justification For Why Not Lower Score:**

N/A

---

### Decision · Program_Chairs · 2024-01-16

Reject